# Safety and Efficacy of PCSK9 Inhibitors in Patients with Acute Coronary Syndrome Who Underwent Coronary Artery Bypass Grafts: A Comparative Retrospective Analysis

**DOI:** 10.3390/jcm13030907

**Published:** 2024-02-04

**Authors:** Giuseppe Nasso, Claudio Larosa, Francesco Bartolomucci, Mario Siro Brigiani, Gaetano Contegiacomo, Maria Antonietta Demola, Walter Vignaroli, Alessandra Tripoli, Cataldo Girasoli, Rosanna Lisco, Marialisa Trivigno, Roberto Michele Tunzi, Tommaso Loizzo, Dritan Hila, Rosalba Franchino, Vincenzo Amodeo, Simone Ventra, Giuseppe Diaferia, Giacomo Schinco, Felice Eugenio Agrò, Maddalena Zingaro, Isabella Rosa, Roberto Lorusso, Armando Del Prete, Giuseppe Santarpino, Giuseppe Speziale

**Affiliations:** 1Department of Cardiac Surgery, Anthea Hospital, GVM Care & Research, 70124 Bari, Italy; msirobrigiani@gvmnet.it (M.S.B.); gcontegiacomo@gvmnet.it (G.C.); mariaantoniettademola@gmail.com (M.A.D.); vignaroli.walter@gmail.com (W.V.); alessandra.tripoli@gmail.com (A.T.); cataldo.girasoli@gmail.com (C.G.); rosannalisco@gmail.com (R.L.); marialisa.trivigno@gmail.com (M.T.); robertotunzi90@gmail.com (R.M.T.); tloizzo@gvmnet.it (T.L.); dritan.hila@gmail.com (D.H.); rfranchino@gvmnet.it (R.F.); simventra@virgilio.it (S.V.); gspeziale@gvmnet.it (G.S.); 2Department of Cardiology, Hospital of Andria, 76123 Andria, Italy; larosa.cld@gmail.com (C.L.); f.bartolomucci@aslbat.it (F.B.); madda.zingaro@gmail.com (M.Z.); rosa.isabella5@gmail.com (I.R.); 3Department of Cardiology, Hospital of Polistena, 89024 Polistena, Italy; enzoamodeo55@libero.it; 4Department of Cardiology, “M. Di Miccoli” Hospital, 70051 Barletta, Italy; gdiaferia@alice.it; 5Health Management, Anthea Hospital, GVM Care & Research, 70124 Bari, Italy; gschinco@gvmnet.it; 6Department of Anesthesiology, University Campus Bio Medico, 00128 Rome, Italy; f.agro@unicampus.it; 7Cardio-Thoracic Surgery Department, Heart and Vascular Centre, Maastricht University, 6229 Maastricht, The Netherlands; roberto.lorusso@mumc.nl; 8Department of Cardiology, Santa Maria Goretti Hospital, 04100 Latina, Italy; armando.delprete85@gmail.com; 9Department of Clinical and Experimental Medicine, Magna Graecia University, 88100 Catanzaro, Italy; gsantarpino@gvmnet.it; 10Department of Cardiac Surgery, Città di Lecce Hospital, GVM Care & Research, 73100 Lecce, Italy

**Keywords:** acute coronary syndrome, evolocumab, statin, revascularization

## Abstract

**Background**. The in-hospital reduction in low-density lipoprotein cholesterol (LDL-C) levels following acute coronary syndrome (ACS) is recommended in the current clinical guidelines. However, the efficacy of proprotein convertase subtilisin–kexin type 9 (PCSK9) inhibitors in those patients undergoing coronary artery bypass graft (CABG) has never been demonstrated. **Methods**. From January 2022 to July 2023, we retrospectively analyzed 74 ACS patients characterized by higher LDL-C levels than guideline targets and who underwent coronary bypass surgery. In the first period (January 2022–January 2023), the patients increased their statin dosage and/or added Ezetimibe (Group STEZE, 43 patients). At a later time (February 2023–July 2023), the patients received not only statins and Ezetimibe but also Evolocumab 140 mg every 2 weeks starting as early as possible (Group STEVO, 31 patients). After one and three months post-discharge, the patients underwent clinical and laboratory controls with an evaluation of the efficacy lipid measurements and every adverse event. **Results**. The two groups did not differ in terms of preoperative risk factors and Euroscore II (STEVO: 2.14 ± 0.75 vs. STEZE: 2.05 ± 0.6, *p* = 0.29). Also, there was no difference between the groups in terms of ACS (ST-, Instable angina, or NSTE) and time of symptoms onset regarding total cholesterol, LDL-C, and HDL-C trends from the preprocedural period to 3-month follow-up, but there was a more significant reduction in LDL-C and total cholesterol in the STEVO group (*p* = 0.01 and *p* = 0.04, respectively) and no difference in HDL-C rise (*p* = 0.12). No deaths were reported. In three STEZE group patients, angina recurrence posed the need for percutaneous re-revascularization. No STEVO patients developed significant adverse events. The statistical difference in these serious events, 7% in STEZE vs. 0% in STEVO, was not significant (*p* = 0.26). **Conclusions**. Evolocumab initiated “as soon as possible” in ACS patients submitted to CABG with high-intensity statin therapy and Ezetimibe was well tolerated and resulted in a substantial and significant reduction in LDL-C levels at discharge, 1 month, and 3 months. This result is associated with a reduction but without a statistical difference between groups.

## 1. Introduction

The reduction in low-density lipoprotein cholesterol (LDL-C) leads to fewer cardiac risks in patients with atherosclerosis and cardiovascular disease [1,2]. Moreover, a favorable biological effect of statins not only on LDL-C reduction but also on inflammation, endothelial function, and coagulation has been postulated [3], particularly during the first period after acute coronary syndrome [4]. These acute coronary syndrome patients are at increased risk of recurrent ischemic events [5]. Consequently, the current clinical guidelines recommend the in-hospital initiation of high-intensity statin treatment following acute coronary syndrome. [6,7]. In this field, some studies have suggested the pleiotropic effects of the protein convertase subtilisin/kexin type 9 (PCSK9) inhibitors. Through various mechanisms, these were confirmed to enhance cardiac function and ischemia-reperfusion injury in animals. Evolocumab reduced major cardiovascular events in secondary prevention [8,9,10,11,12]. The feasibility, safety, and efficacy of PCSK9 antibody treatment during acute coronary syndrome in patients who have undergone CABG are presently unknown [13]. The only information we have, but only in patients with ACS not submitted to CABG, demonstrated that Evolocumab started in-hospital on top of high-intensity statin treatment was well tolerated and resulted in a substantial reduction in LDL-C levels after 8 weeks [14].

Moreover, ischemic and reperfusion myocardial injury is often reported in patients who have undergone CABG [15,16,17], and it is associated with critical complications [18]. The mechanisms caused by ischemic and reperfusion myocardial injury during surgery are complex, and include thrombosis, oxygen-free radicals, inflammation, endothelial dysfunction, changes in cellular metabolism, and immune reactions [19]. Many clinical trials have been conducted to decrease myocardial injury during CABG surgery. However, statins are currently the only treatment approved for clinical use [20,21,22].

In this field, PCSK9 antibodies are also considered cardioprotective due to their anti-inflammatory, antioxidant, and immunomodulatory effects [23,24,25]. Indeed, in the field of CABGs, the PCSK9 inhibitor’s pleiotropic effects have to be considered. The anti-inflammatory properties are attributed to the distribution on arterial walls, which promotes inflammation and atherosclerosis [26]. Furthermore, studies on animal models have indicated that PCSK9 inhibitors exhibit potential immunoregulatory properties, whereas septic shock ones were shown to benefit from PCSK9 inhibitor administration [27]. Moreover, it was reported that it reduces ischemic and reperfusion damage of the myocardium before administration, thus improving cardiac function [28].

Given its potential cardioprotective effects, it can be speculated that PCSK9 inhibitors would be helpful for patients undergoing CABG. However, this has not been demonstrated. There is only one ongoing trial on the effect of PCSK9 inhibitors following bypass surgery in the general population, and the results are not yet known [29]. 

Moreover, as the recent 2023 European guidelines highlight, the optimal timing of PCSK9 inhibitor treatment initiation remains to be determined [30].

We present our retrospective experience using Evolocumab associated with statins in ACS patients undergoing CABG surgery compared with patients receiving only high-dosage statins and Ezetimibe.

## 2. Methods

### 2.1. Ethical Statement

The present study conforms to the Declaration of Helsinki. It was approved by our institutional review board for human research for retrospective studies (Prot. Number. 101.10.2023) and registered in 21 October 2023 (I.D. study 781; code number AH.011.02.023) to the observational registry for clinical trials of the “Italian Agency of Medicines” (AIFA—Agenzia Italiana del Farmaco). Patients gave consent to use their clinical data for research purposes.

### 2.2. Study Population

From January 2022 to July 2023, we retrospectively analyzed all patients presenting with ACS whose LDL-C levels were above established limits and who underwent coronary bypass surgery. 

In particular, we observed the patients with ST- and non-ST-segment elevation acute coronary syndrome (STE and NSTE-ACS) with a symptom onset of ≤72 h and clinical/hemodynamic stability, usually admitted to our cardiac surgery department from other cardiological/catheter laboratories of the regional spoke hospitals.

Total cholesterol, HDL-C, and LDL-C were assessed for every patient at admission. From January 2022 to January 2023, every patient with the described characteristics and LDL ≥ 100 mg/dL received the following protocol as a routine clinical practice:(1)If without statins, they received a moderate-intensity statin (e.g., 40 mg) from the catheter laboratory (cath lab) table up to the 3-month follow-up;(2)On statins, they added Ezetimibe and a higher statin dosage (80 mg) from the cath lab table up to the 3-month follow-up.

Following the new scientific suggestions [31,32] from February 2023 for patients with LDL ≥ 100 mg/dL, the routine treatment protocol was changed as follows:(1)If without statins, they received a moderate-intensity statin (e.g., 40 mg) + Ezetimibe + PCSK9i (Evolocumab 140 mg every 2 weeks) from the cath lab table up to the 3-month follow-up;(2)On statins, they added Ezetimibe and a higher statin dosage (80 mg) + PCSK9i (Evolocumab 140 mg every 2 weeks) from the cath lab table up to the 3-month follow-up.

The PCSK9i was administered at baseline “as early as possible,” usually before the coronary angiography. 

In the follow-up after discharge, an adjustment on dosage of statin therapy and/or an addition of non-statin lipid-lowering therapies were recorded.

One and three months after discharge, as per institutional protocol, the patients underwent a clinical and laboratory control to evaluate the efficacy of lipid measurements, including total cholesterol, triglycerides, HDL cholesterol, and LDL cholesterol. 

Every adverse event, in particular, death, re-myocardial infarction, coronary re-revascularization, re-hospitalization for recurrent ACS, hospitalization for heart failure, and cerebrovascular event, was recorded up to the last follow-up at 3 months.

All patients of the STEVO group were submitted to coronary tomography; however, this parameter was not used as an endpoint in the study.

### 2.3. Statistical Analysis

Baseline characteristics were compared using Student’s *t*-tests, Fisher exact tests, and chi-square tests. For safety outcomes, missing data were not imputed, and we summarized adverse events dividing by treatment group through descriptive statistics with rate ratios from Mantel–Cox regression (first event of each type) or Poisson regression (number of events of each type), with the time-at-risk equivalent dated at the 3-month follow-up or death. The distribution of the single variables to demonstrate a normal distribution was performed following the visualization of the variable into the histogram. Tests were 2-sided, and a *p*-value < 0.05 was considered significant. Analyses were performed with SPSS 12.0 (IBM Corp., Armonk, NY, USA).

## 3. Results

In the study period, we had 74 patients with the requested characteristics; in particular, from January 2022 to January 2023, 43 patients underwent statin 40 or 80 + Ezetimibe (called Group STEZE), and from February 2023 to July 2023, 31 patients underwent statin 40 + Evolocumab or statin 80 + Ezetimibe + Evolocumab (called Group STEVO). 

The baseline patient characteristics are shown in Table 1. The two groups did not differ in terms of risk factors as to mean age (STEVO: 68.5 ± 6.8 y.o. vs. STEZE: 70 ± 7.4 y.o. *p* = 0.38), diabetes (STEVO: 12–38.7% vs. STEZE: 18–41.8%, *p* = 0.79), previous myocardial infarction (STEVO: 7–22.6% vs. STEZE: 10–23.2%, *p* = 0.91), left ventricular ejection fraction (STEVO: 51 ± 11% vs. STEZE: 48 ± 14% *p* = NS), and Euroscore II (STEVO: 2.14 ± 0.75 vs. STEZE: 2.05 ± 0.6, *p* = 0.29). Also, in terms of ACS (ST-, Instable angina, or NSTE) and time of symptom onset, there was no difference between the groups (Table 1). Most patients (74.4% in STEZE and 71% in STEVO, *p* = 0.88) were not on statin at baseline (Table 2). Mean baseline-calculated LDL-C levels were 158 ± 14 mg/dL in STEVO and 154 ± 18 mg/dL in STEZE (*p* = 0.72). Also, total cholesterol (STEVO: 195 ± 13 mg/dL vs. STEZE: 190 ± 17 mg/dL, *p* = 0.66) and HDL-C (STEVO: 37 ± 4 mg/dL vs. STEZE: 36 ± 5 mg/dL, *p* = 0.89) were high, and did not differ between groups. Figure 1 shows the cholesterol, LDL-C, and HDL trends from the preprocedural period to 3-month follow-up in both groups, demonstrating a more significant reduction in LDL-C and cholesterol in STEVO (*p* = 0.01 and *p* = 0.04, respectively) and no difference in HDL-C raising (*p* = 0.12) (see Figure 1). 

The reduction in LDL-C values was evident after 4 weeks, and was also maintained at 12 weeks in both groups (Figure 1). Three patients in STEVO (11.5%) and four in STEZE (11.8%) underwent primary PCI on the culprit lesion and, after that, received a surgical coronary completion within 72 h. We achieved in all patients a complete revascularization with a mean number of anastomoses of 3.3 ± 1 in STEZE and 3.6 ± 1 in STEVO. All patients received revascularization of the left anterior coronary artery with the left internal mammary artery, and the other bypasses with the saphein vein using the sequential technique. There were no differences in type of revascularization or surgical techniques between the STEVO and STEZE groups. Two STEZE patients and one STEVO patient underwent off-pump surgery. The 71 remaining patients underwent cardiopulmonary bypass (CPB) and cross-clamp (X-Cl) without differences between the groups (CPB: 78.9 ± 22 min in STEZE and 84 ± 18 min in STEVO, *p* = 0.22; X-Cl: 54.9 ± 11 min in STEZE and 61 ± 17 min in STEVO, *p* = 0.16). No differences were recorded in terms of the number of bypass grafts (2.5 ± 1.1 bypass in STEZE and 2.6 ± 1.2 bypass in STEVO, *p* = 0.40). One patient in the STEVO group did not receive the Evolocumab before the coronary angiography, but received it immediately after that. The hospitalization length was 12 ± 3 days in STEVO vs 11.8 ± 3 days in STEZE (*p* = 0.18), and the ICU stay was 2.8 ± 1 days in STEVO vs. 3.1 ± 1 days in STEZE (*p* = 0.27). Two patients in both groups (7.7% and 5.9%) needed a surgical revision due to bleeding (*p* = 0.09). The visits at 4 and 12 weeks were performed in all 74 patients as routinely scheduled for each patient submitted to CABG in our institution. The lipid measurements are shown in Table 3. Musculoskeletal pain was the most reported adverse event (15 patients from all populations, 20.3%). In particular, of six STEVO patients (19.3%) with consequences, three received statin reductions (from 80 to 40 mg) and three received statin suspensions; of nine STEZE patients (20.9%) with consequences, six received reductions from 80 mg to 40 mg and three received suspensions. Other common events were diarrhea (in three STEVO patients, 9.7%) and local injection site reaction (in five STEVO patients, 16.1%). The Alanine Aminotransferease (ALT) increased in both groups up to 3 months (see Table 3), but in no patient increased more than three times the upper limit of normal. Adverse events that led to Evolocumab discontinuation did not occur. No deaths were reported. In terms of the adverse events, up to 3 months, neither re-myocardial infarction nor cerebrovascular events were recorded in either group. However, in three STEZE group patients, a recurrence of angina posed the need for re-hospitalization and angiography, demonstrating in both cases a venous graft thrombosis and consequent percutaneous coronary re-revascularization of the native vessels. No adverse events occurred in STEVO patients. The statistical difference in these serious events, 7% in STEZE vs. 0% in STEVO, was not significant (*p* = 0.26), but a trend in favor of STEVO was detected. Mean levels of CRP decreased from baseline to 4 and 12 weeks without significant differences between groups, as did the levels of Troponin I up to discharge, as shown in Table 3. Between 1 and 3 months, every STEVO patient underwent Coro-CT as screening, and all the bypasses were identified and described as functioning (Figure 2).

## 4. Discussion

It has been demonstrated that, through various mechanisms, PCSK9 inhibition improves cardiac function and ischemia-reperfusion injury in animal models. As a consequence, a protective effect of PCSK9 inhibitors in the clinical field, because ischemia-reperfusion injury is inevitable in these patients, is predictable. Usually, a cardiac enzyme elevation related to this myocardial injury occurs in almost all patients undergoing CABG in the first postoperative period. Moreover, a perioperative myocardial infarction may occur, e.g., due to graft failure, early after CABG surgery. In our study, we recorded the routine measurement of cardiac enzymes at 24 h and 72 h after surgery to monitor possible perioperative myocardial infarction (MI). We cannot demonstrate a cardioprotective benefit of Evolocumab because there are no differences in enzyme elevation in either group.

However, we have to highlight that our study is the first clinical and laboratory analysis of Evolocumab treatment initiated in-hospital in patients presenting with STE- and NSTE-ACS who underwent urgent surgical revascularization; nonetheless, we did not detect an immediate difference in the cardioprotective effect.

On the other hand, we showed that the addition of Evolocumab 140 mg as early as possible during ACS and subsequently every 2 weeks to high-intensity statins (STEVO group), compared with high-intensity statins and Ezetimibe (STEZE), resulted in a significant reduction in LDL-C levels after 3 months. A limitation of this study is the limited number of patients, although they were treated in a short period of time. However, we highlight that the treatment was well tolerated, without significant imbalances in adverse events. A discrete increment in the ALT value at 1 and 3 months was recorded in both groups. Our approach with Evolocumab “fast track” opposes the “stepwise” approach consisting of the early start of statin therapy, followed by the subsequent addition of Ezetimibe, and Evolocumab treatment only at the end if the LDL-C levels remain elevated [4,5]. The negative side of the “stepwise” approach is that ACS patients with markedly elevated LDL-C levels received the Evolocumab several months following their index event, while the early period after ACS is the period most at risk of recurrent ischemic events [1]. In fact, a precocious initiation of a high dosage of statin therapy following an ACS has been shown to reduce the occurrence of early recurrent events [6,17]. This is the reason why the EVOPACS study tested the efficacy of the in-hospital start of Evolocumab in patients who had uncontrolled LDL-C levels, despite pre-existing high-dosage statin treatment during ACS [12].

On the other hand, in our study, we demonstrated, in all the patients who underwent CABG, not only the reduction in LDL-C but also the clinical outcome at 3 months of an advantage in the STEVO group with less need of repeated revascularization. Our results build upon previous studies that investigated Evolocumab in ACS [8,12], patients with statin intolerance [19] or familial hypercholesterolemia [9], and patients with stable manifestations of ischemic cardiomyopathy [10,11]. In particular, the FOURIER trial showed the efficacy of Evolocumab in significantly reducing the risk of cardiac and vascular events in patients after ASC [11], with greater risk reduction observed in patients closer to their index MI; particularly, the median interval between index MI and study enrollment ranged from 4 months up to 11 years in FOURIER [12]. Similarly, the ODYSSEY OUTCOMES trial evaluated the cardiovascular effects of alirocumab in patients at least 1 month (median 2.6 months) after ACS [15]. One of the original points of our study is the administration of Evolocumab very early, preferably before the coronary angiography. The mean 40.7% LDL-C decrease achieved with Evolocumab versus placebo in EVOPACS, compared with approximately 60% in prior Evolocumab trials [8,9,10,11,18], should be read in light of important differences regarding background statin therapy. As in the EVOPACS, we also initiated Evolocumab in those who were not taking statins at baseline; in fact, as a retrospective real-life analysis, this reflects the majority of ACS patients [18,20].

Moreover, the characteristics of our population also justify the reduction in LDL-C in the STEZE group. These results are coherent with safety and tolerability data highlighted in previous studies with Evolocumab in stable clinical settings. Our results showed that the rate of adverse cardiovascular events was numerically higher in the STEZE group (7% versus 0%) but did not contrast significantly with the STEVO group and was like that found in ACS trials. These results need to be read in view of the small sample size as well as the inclusion of broadly representative ACS patients with a lot of comorbidities and with a short follow-up. However, a very important numerical difference (7% vs. 0%) in the need for repeated revascularization could demonstrate an important effect of Evolocumab on such events in the early post-ACS period and merits further investigation with a longer follow-up in order to emphasize this difference. Our study, in fact, clearly reports an advantageous trend in terms of no-repeat revascularization in the STEVO group.

Moreover, we found that every STEVO patient up to 3 months had functioning bypasses, demonstrated with Coro-CT. This is a demonstration of no STEVO patients having a silent bypass dysfunction. Also, the CRP level analysis demonstrates in our study a neutral effect of Evolocumab, confirming the results of previous studies [11,29]. At the same time, it is surprising to us. In fact, we speculated about the PCSK9 inhibitors and their benefits for patients undergoing cardiovascular surgery. Based on animal studies, the PCSK9 inhibitors exhibited potential immunoregulatory properties [27].

Moreover, it demonstrated an anti-inflammatory effect due to its distribution on arterial vascular walls [26]. In summary, the anti-inflammatory effect of PCSK9 inhibitors in patients undergoing cardiovascular surgery has not been demonstrated in our study, probably due to the short-term follow-up and a numerically small study population. 

### Study Limitations

The retrospective analysis is the first limitation, while nothing is known and published, to the best of our knowledge, from an urgent surgical population. However, a prospective randomized trial is mandatory to confirm our results. Moreover, although Evolocumab reduces LDL-C levels rapidly (within days) [25], the last measurement of lipid levels was 3 months after the first administration; thus, we could not capture the long-term effects of Evolocumab in our population. We will continue to follow up on this population in the future to resolve that limitation. However, the importance of our preliminary findings is maintained, notwithstanding the short-term follow-up. We have the CT-scan controls only for the Evolocumab patients, which is a limitation from our point of view: if there is a difference in bypass patency at discharge between groups, there may be a difference in subsequent follow-up complications, repeated revascularizations, etc. But, without a control CT scan, in our study, this can be only speculated. Moreover, we have to highlight that our study lacks information about treatment-dependent changes in Lp(a) and a potential difference linked to gender. In fact, there are studies that demonstrated the presence of differences in response to treatment with PCSK9i between men and women, and these are essential to gain a better understanding of the relationship between LDL-C and Lp(a) lowering in response to PCSK9i [33]. 

Given the limited number of patients and the short follow-up period, we underline again that our study is a preliminary investigation, and the results must therefore be confirmed by an analysis on a larger sample.

## 5. Conclusions

In patients presenting with ACS who underwent CABG, Evolocumab initiated “as soon as possible” with a high dosage of statin therapy and Ezetimibe was well tolerated and resulted in a substantial and significant reduction in LDL-C levels at dismission, 1 month, and 3 months compared with only statins + Ezetimibe. This laboratory result is associated with a concomitant reduction trend in repeated revascularization in the STEVO group but without a statistical difference between groups.

## Figures and Tables

**Figure 1 jcm-13-00907-f001:**
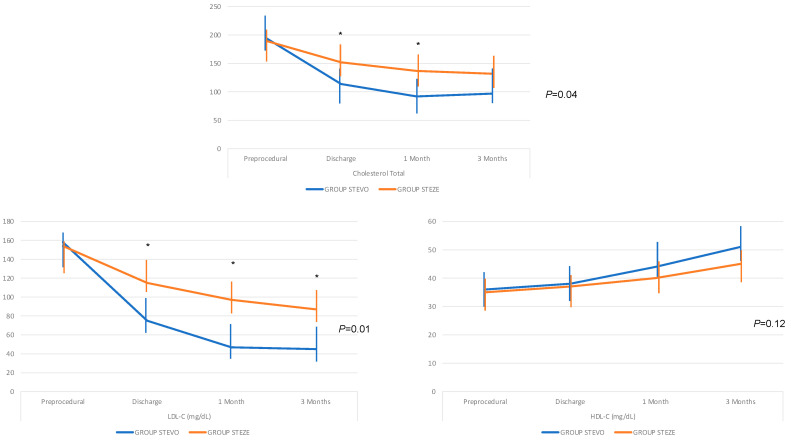
Total cholesterol, LDL-C, and HDL-C trends. “*” represents all time points where *p* was <0.05.

**Figure 2 jcm-13-00907-f002:**
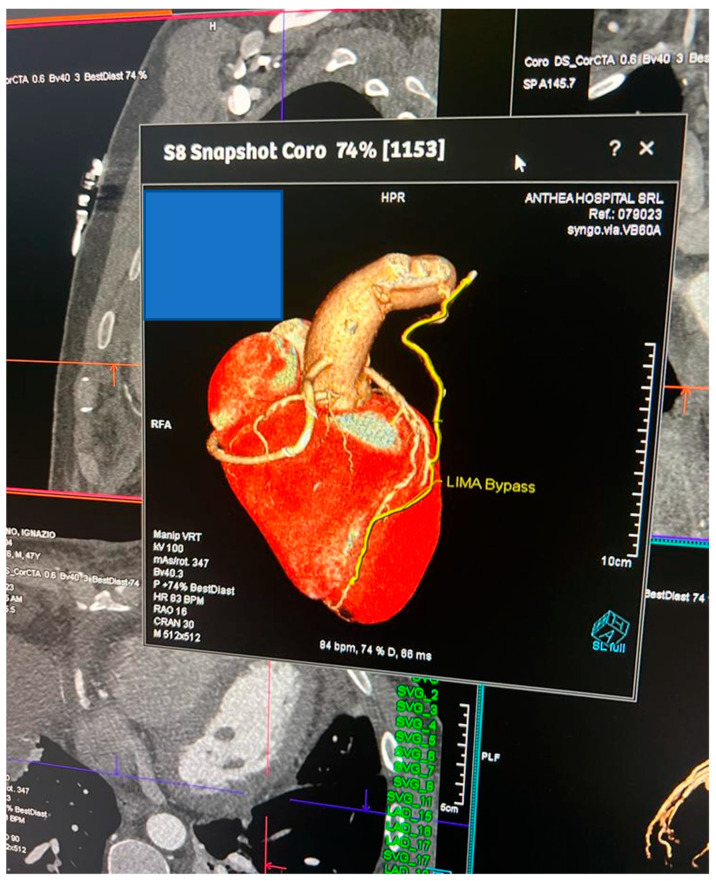
Example of a picture from Coro-CT in a STEVO patient.

**Table 1 jcm-13-00907-t001:** Patients’ characteristics.

	GROUP STEVO	GROUP STEZE	*p*
	pts = 31	pts = 43	
Age, yrs	68.5 ± 6.8	70 ± 7.4	NS
Male	26 (83.9%)	34 (79.1%)	NS
Body mass index, kg/m^2^	26.9 ± 3	26.4 ± 2.9	NS
Diabetes mellitus	12 (38.7%)	18 (41.8%)	NS
Insulin-treated	7 (22.6%)	10 (23.2%)	NS
Arterial hypertension	21 (67.7%)	30 (69.8%)	NS
Active smoking	19 (61.3%)	25 (58.1%)	NS
Previous myocardial infarction	7 (22.6%)	10 (23.2%)	NS
Previous PCI	15 (48.4%)	20 (46.5%)	NS
Previous CABG	0	0	
Peripheral arterial disease	2 (6.4%)	4 (9.3%)	NS
History of stroke	0	2 (4.6%)	NS
History of TIA	1 (3.2%)	2 (4.6%)	NS
History of malignancy	6 (19.3%)	8 (18.6%)	NS
Euroscore II	2.14 ± 0.75	2.05 ± 0.6	NS
Time of symptom onset, h	98 ± 30	91 ± 27	NS
Culprit primary PCI	4 (12.9%)	5 (11.6%)	NS
NSTE-ACS	21 (67.7%)	30 (69.8%)	NS
STE-ACS	4 (12.9%)	5 (11.6%)	NS
Instable angina	6 (19.3%)	8 (18.6%)	NS

PCI: percutaneous coronary intervention; CABG: coronary artery bypass grafting; TIA: transitory ischemic attack; NSTE: non-ST elevation; STE: ST elevation; ACS: acute coronary syndrome; NS: not significant.

**Table 2 jcm-13-00907-t002:** Pre-ACS statin treatment.

GROUP STEZE					GROUP STEVO				
*January 2022–* *January 2023*	*No Statin*	*Low- or* *Moderate-* *Intensity Statin*	*High-* *Intensity Statin*	*Ezetimibe Treatment*	*February 2023–June 2023*	*No Statin*	*Low- or* *Moderate-* *Intensity Statin*	*High-* *Intensity Statin*	*Ezetimibe Treatment*
Pt. 1	x				Pt. 1		x		
Pt. 2	x				Pt. 2	x			
Pt. 3	x				Pt. 3	x			
Pt. 4		x			Pt. 4	x			
Pt. 5	x				Pt. 5	x			
Pt. 6	x				Pt. 6	x			
Pt. 7	x				Pt. 7	x			
Pt. 8	x				Pt. 8	x			
Pt. 9			X		Pt. 9	x			
Pt. 10		x			Pt. 10	x			
Pt. 11	x				Pt. 11		x		
Pt. 12	x				Pt. 12			x	
Pt. 13				x	Pt. 13	x			
Pt. 14	x				Pt. 14				x
Pt. 15	x				Pt. 15	x			
Pt. 16	x				Pt. 16	x			
Pt. 17	x				Pt. 17		x		
Pt. 18	x				Pt. 18	x			
Pt. 19	x				Pt. 19			x	
Pt. 20			X		Pt. 20		x		
Pt. 21		x			Pt. 21	x			
Pt. 22	x				Pt. 22	x			
Pt. 23	x				Pt. 23	x			
Pt. 24	x				Pt. 24	x			
Pt. 25			X		Pt. 25	x			
Pt. 26		x			Pt. 26	x			
Pt. 27		x			Pt. 27	x			
Pt. 28	x				Pt. 28	x			
Pt. 29	x				Pt. 29	x			
Pt. 30	x				Pt. 30			x	
Pt. 31	x				Pt. 31		x		
Pt. 32	x					71%	16.1%	9.7%	3.2%
Pt. 33	x								
Pt. 34	x								
Pt. 35	x								
Pt. 36	x								
Pt. 37	x								
Pt. 38	x								
Pt. 39		x							
Pt. 40			X						
Pt. 41	x								
Pt. 42	x								
Pt. 43	x								
	74.4%	14%	9%	2.6%					

**Table 3 jcm-13-00907-t003:** Laboratory.

	Total Cholesterol (mg/dL)				
	Preprocedural	Discharge	1 Month	3 Months	*p*
GROUP STEVO	195 ± 13	114 ± 16	92 ± 12	97 ± 9	0.04
GROUP STEZE	190 ± 17	152 ± 12	137 ± 16	132 ± 17	
	LDL-C (mg/dL)				
	Preprocedural	Discharge	1 Month	3 Months	*p*
GROUP STEVO	158 ± 14	75 ± 16	47 ± 10	45 ± 10	0.01
GROUP STEZE	154 ± 18	115 ± 11	97 ± 19	87 ± 22	
	HDL-C (mg/dL)				
	Preprocedural	Discharge	1 Month	3 Months	*p*
GROUP STEVO	36 ± 4	38 ± 3	44 ± 5	51 ± 6	0.12
GROUP STEZE	35 ± 5	37 ± 4	40 ± 5	45 ± 9	
	Triglyceride (mg/dL)				
	Preprocedural	Discharge	1 Month	3 Months	*p*
GROUP STEVO	176 ± 17	166 ± 15	141 ± 27	133 ± 22	0.18
GROUP STEZE	179 ± 20	173 ± 17	170 ± 16	173 ± 16	
	Alanine Aminotransferease (ALT) (U/L)				
	Preprocedural	Discharge	1 Month	3 Months	*p*
GROUP STEVO	37 ± 5	45 ± 9	62 ± 22	58 ± 24	0.44
GROUP STEZE	35 ± 2	39 ± 6	59 ± 22	57 ± 20	
	Troponin I (microgr/L)				
	Preprocedural	24 h	72 h	Discharge	*p*
GROUP STEVO	613 ± 257	829 ± 323	364 ± 109	138 ± 66	0.06
GROUP STEZE	613 ± 273	821 ± 316	507 ± 218	298 ± 184	
	CRP (mg/dL)				
	Preprocedural	Discharge	1 Month	3 Months	*p*
GROUP STEVO	52 ± 31	24 ± 15	3.5 ± 2	3.1 ± 2	0.10
GROUP STEZE	64 ± 58	64 ± 58	6.9 ± 3	5.8 ± 3	

## Data Availability

The original contributions presented in the study are included in the article, further inquiries can be directed to the corresponding author.

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
