# Peer review of "Safety and Efficacy of PCSK9 Inhibitors in Patients with Acute Coronary Syndrome Who Underwent Coronary Artery Bypass Grafts: A Comparative Retrospective Analysis"

_jcm, 2024, doi:10.3390/jcm13030907_

Round 1
Reviewer 1 Report
Comments and Suggestions for Authors
Dear Editor,
I carefully read the manuscript "Safety and Efficacy of PCSK9 inhibitor in patients with acute coronary syndrome who underwent to Coronary Artery Bypass Graft: a comparative retrospective analysis".
My comments and suggestions for the authors are the following:
- The abstract is verbose. It is difficult to focus on important information among the many useless ones. Moreover, in the abstract, the authors wrote HDL referring to HDL-C. Please, revise accordingly.
- In the abstract, the authors wrote "Cholesterol, LDL-C and HDL trends from preprocedural to 3 months follow up in both groups". What do they refer to with the word "cholesterol"?
- Ref. 23 and 24 refers to Alirocumab and not to statins. In effect, the anti-inflammatory effect following statin treatment has only been demonstrated acutely and is still a matter of discussion among lipidologists. For this reason, this sentence should be reworded and the two references removed since are out of context, here.
- The hypothetical anti-inflammatory effects following PCSK9 inhibition should also be discussed more extensively. In effect, treatment with PCSK9 inhibitors has been shown not to decrease hs-CRP levels.
- Overall, English language needs to be carefully revised and improved.
- As regards statistical analysis, the authors should specify how the normal distribution of the variables was assessed.
- In the methods, the authors should further specify the assessments they performed.
- In Table 1, I warmly suggest to the authors to include p-values only when statistically significant.
- Information reported in Table 2 should more properly pooled and included in Table 1. This would significantly improve the readability of the manuscript. However, I suggest to the authors to upload the information currently reported in Table 2 as Supplementary Material.
- Among the study's limitations, I suggest to the authors to refer also to the lack of information about treatment-dependant change in Lp(a). In this regard, they should refer to the recently published study doi: 10.3390/biomedicines11123271.
Comments on the Quality of English LanguageEnglish language needs to be carefully revised by a native English speaking person.
Author Response
My comments and suggestions for the authors are the following:
1) The abstract is verbose. It is difficult to focus on important information among the many useless ones. Moreover, in the abstract, the authors wrote HDL referring to HDL-C. Please, revise accordingly.
1) THANK YOU FOR YOUR SUGGESTIONS, WE REVISED/SEMPLIFIED THE ABSTRACT FOLLOWING YOUR OPINION.
2) In the abstract, the authors wrote "Cholesterol, LDL-C and HDL trends from preprocedural to 3 months follow up in both groups". What do they refer to with the word "cholesterol"?
2) WE ADDED “TOTAL” TO CHOLESTEROL TO EXPLAIN THE MEANING.
- Ref. 23 and 24 refers to Alirocumab and not to statins. In effect, the anti-inflammatory effect following statin treatment has only been demonstrated acutely and is still a matter of discussion among lipidologists. For this reason, this sentence should be reworded and the two references removed since are out of context, here.
3) WE MODIFIED THE SENTENCE REFERING IT TO PCSK9 ANTIBODIES.
4) The hypothetical anti-inflammatory effects following PCSK9 inhibition should also be discussed more extensively. In effect, treatment with PCSK9 inhibitors has been shown not to decrease hs-CRP levels.
4) AS SUGGESTED, WE DISCUSSED MORE EXTENSIVELY THE HYPOTHTICAL ANTI-INFLAMMATORY EFFECT AND WE HIGHLIGHT THAT WE ARE NOT ABLE TO DEMONSTRATE IT WITH OUR STUDY.
5) Overall, English language needs to be carefully revised and improved.
5) ENGLISH LANGUAGE WAS CAREFULLY REVISED BY A NATIVE ENGLISH SPEAKING PERSON.
6) As regards statistical analysis, the authors should specify how the normal distribution of the variables was assessed.
6) WE SPECIFIED THE VISUALIZATION INTO THE HISTOGRAM
7) In the methods, the authors should further specify the assessments they performed.
7) AS RETROSPECTIVE STUDY, WE DESCRIBED THE INSITUTIONAL PROTOCOL FOR THE MONITORING OF THE PATIENTS OPERATED. THERE IS NOT A “STUDY PROTOCOL” BUT “AN INSTITUTIONAL PROTOCOL” BECAUSE IT IS NOT A PROSPECTIVE TRIAL.
8) In Table 1, I warmly suggest to the authors to include p-values only when statistically significant.
8) AS WARMLY SUGGESTED, WE USED “NOT SIGNIFICANT” FOR THE P>0.05.
9) Information reported in Table 2 should more properly pooled and included in Table 1. This would significantly improve the readability of the manuscript. However, I suggest to the authors to upload the information currently reported in Table 2 as Supplementary Material.
9) WE NOW TRANSFORM THE TABLE 2 AS APPENDIX 1
10) Among the study's limitations, I suggest to the authors to refer also to the lack of information about treatment-dependant change in Lp(a). In this regard, they should refer to the recently published study doi: 10.3390/biomedicines11123271.
10) WE ADDED THIS LIMITATION AND THE REQUESTED REFERENCE.

Reviewer 2 Report
Comments and Suggestions for Authors
This study reported the 12-week lipid profile and the presence of major cardiovascular events (MACE) in CABG patients treated with lipid-lowering oral therapy plus a PCKS-9 inhibitor. While of interest, there are several issues.
#1) It is a retrospective study, but it is unclear why the authors chose a 12-week study period; it would seem natural that a PCSK-9 inhibitor would lower the lipid profile early in the study. Therefore, it is unlikely that there would be a difference in MACE at 12 weeks. Since this is a retrospective study, it is possible that a difference could be seen if the study was conducted at 24 weeks.
#2 The authors should indicate the left ventricular ejection fraction at the enrollment and/or discharge and the number of bypass grafts between the two groups.
#3 Normally, a cardiac CT scan is performed at discharge to see if the bypass is still open, but it seems to be performed only in the PCSK-9 inhibitor group but not in the medical group. If there is a difference in bypass patency at discharge, there may be a difference in subsequent MACE.
#4 The authors conclude that there was a trend toward a difference in MACE, but it is not statistically significant. The authors should state this accurately.
Author Response
#1) It is a retrospective study, but it is unclear why the authors chose a 12-week study period; it would seem natural that a PCSK-9 inhibitor would lower the lipid profile early in the study. Therefore, it is unlikely that there would be a difference in MACE at 12 weeks. Since this is a retrospective study, it is possible that a difference could be seen if the study was conducted at 24 weeks.
1) AS RETROSPECTIVE STUDY, WE DO NOT CHOOSE THE FOLLOW UP TIME, WE USED THE “REGULAR” INSTITUTIONAL FOLLOW UP VISIT. WE ALSO AGREE WITH THE REVIEWER AND WE’LL GO AHEAD WITH FUTURE CONTROLS TO SEE POSSIBLE EVENTS IN A LONGER FOLLOW UP
#2 The authors should indicate the left ventricular ejection fraction at the enrollment and/or discharge and the number of bypass grafts between the two groups.
2) WE ADDED THESE DATA IN THE RESULTS SECTION
#3 Normally, a cardiac CT scan is performed at discharge to see if the bypass is still open, but it seems to be performed only in the PCSK-9 inhibitor group but not in the medical group. If there is a difference in bypass patency at discharge, there may be a difference in subsequent MACE.
3) IT IS A DESCRIPTIVE ASPECT ONLY FOR THE EVOLOCUMAB GROUP, BECAUSE THERE ARE NO CONTROLS IN THE STANDARD GROUP. NOW WE SPECIFIED IT BETTER IN THE LIMITATIONS
#4 The authors conclude that there was a trend toward a difference in MACE, but it is not statistically significant. The authors should state this accurately.
4) WE HIGHLIGHT BETTER THE ABSENCE OF A STATISTICALLY SIGNIFICANCE IN THE CLINICAL RESULTS.

Round 2
Reviewer 1 Report
Comments and Suggestions for Authors
The manuscript has been improved with the revision.
Comments on the Quality of English LanguageThere are still some typos.
Author Response
1) There are still some typos.
1) the paper underwent a second round of an English Language control with a native English speaker.
Reviewer 2 Report
Comments and Suggestions for Authors
The rationale for setting the follow-up period at 12 weeks remains unclear, but the authors have generally corrected the points pointed out.
Please correct the following two points:.
#1 Normally, it should be total cholesterol.
#2 The P-value is shown before and after the following sentence, so please show the P-value in this sentence as well.
No difference were recoded in terms of number of bypass grafts (2.5±1.1 bypass in STEZE and 2.6±1.2 bypass in STEVO, p=NS).
Author Response
The rationale for setting the follow-up period at 12 weeks remains unclear, but the authors have generally corrected the points pointed out.
Please correct the following two points:
1) Normally, it should be total cholesterol.
1) we correct in the paper “total cholesterol”
2) The P-value is shown before and after the following sentence, so please show the P-value in this sentence as well.
No difference were recorded in terms of number of bypass grafts (2.5±1.1 bypass in STEZE and 2.6±1.2 bypass in STEVO, p=NS).
2) we correct the sentence as request